# Effects of Basic Psychological Needs on Physical Self-Efficacy and Attitudes toward PE in Korean Middle-School Physical Education

**DOI:** 10.3390/healthcare12010091

**Published:** 2023-12-30

**Authors:** Jongseub Shin, Heonsu Gwon

**Affiliations:** 1Rehabilitation Sports Research Institute, Korea Nazarene University, 48, Wolbong-ro, Seobuk-gu, Cheonan-si 31172, Chungcheongnam-do, Republic of Korea; jstkd68@gmail.com; 2Industry-Academia Cooperation Center, Yong In University, 134, Yongindaehak-ro, Cheoin-gu, Yongin-si 17092, Gyeonggi-do, Republic of Korea

**Keywords:** middle school, physical education, basic psychological need, physical self-efficacy, physical education attitude, South Korea

## Abstract

This study explored the effect of basic psychological needs in secondary physical education (PE) classes in South Korea on physical self-efficacy and attitudes toward PE. Data from 296 middle-school students were collected from May to June 2022 in Seoul or Gyeonggi-do. Participants were surveyed via simple random sampling. Confirmatory factor, correlation, and multiple regression analyses were conducted, and reliability was assessed with Cronbach’s α. Basic psychological needs had a significant positive effect on physical self-efficacy. Competence and autonomy positively and significantly affected perceived physical competence (β = 0.535 and 0.320, respectively). Basic psychological needs had a significant positive effect on classroom attitudes toward PE. Autonomy and relatedness positively and significantly affected basic attitudes (β = 0.317 and 0.388, respectively) and social attitudes (β = 0.3498 and 0.213, respectively). Physical self-efficacy had a significant positive effect on PE classroom attitudes. Perceived physical competence and physical self-presentation confidence had a positive and significant effect on basic attitudes (β = 0.258 and 0.166, respectively). The results implied that attitudes toward school life can be improved through physical activity in secondary PE classes.

## 1. Introduction

Among various subjects taught at school, physical education (PE) has had a considerably positive effect on students. Most students with positive conceptions of PE have had enjoyable, happy, and exciting experiences therein [1]. Students enjoy PE because it provides them with opportunities to release stress, energy, and endorphins through physical activity [2]. While participating in physical activities during PE, students acquire various physical functions. They also learn cooperation and good sportsmanship derived from this process, thus achieving the goals of PE [3]. As such, participation in physical activities provided during PE affects character-building, as well as physical, intellectual, and emotional development. This enables the acquisition of values, ultimately facilitating their psychological stability.

Moreover, the theory of basic psychological needs—including autonomy, competence, and relatedness—was developed in the field of psychology to verify the personal stability gained through the social environment [4]. Individuals tend to continuously pursue the satisfaction of these three basic psychological needs [5]. Human behavior and intention are determined by the degree to which these three basic psychological needs are fulfilled [6]. As such, the theory of basic psychology requires clarification with respect to the conceptual approach to basic human needs and the basic state of human psychology [4].

When psychological needs such as pleasure, happiness, and stability are satisfied, people become more engaged in physical activities. As a result, it can improve a team’s chance of success and help individuals achieve the goals they want to achieve. However, if psychological needs are not satisfied, it is difficult to expect a positive result [4,7]. Therefore, the satisfaction of basic psychological needs during PE is a variable that can change behaviors and intentions toward physical activity.

In particular, the adolescent period is an intermediate stage during which one moves from childhood into adulthood, and maturation occurs physically, mentally, and socially [8]. During this period, the participation of adolescents in physical activities is vital to their growth into healthy adults [9]. On average, students who engage in physical activity for three days or more per week demonstrate higher satisfaction in life than those who do not engage in any physical activity [10]. Adolescents who frequently participate in exercise can maintain a positive mental state, which could have a positive effect on basic psychological needs [11]. This leads to successful results, as one consequently engages in exercise with confidence.

The confidence felt when exercising can be explained by the following variable: physical self-efficacy. This is the confidence felt when performing exercise tasks related to the body. Gill et al. [12] said that exercise participation improves when using the body with confidence. It was mentioned that if psychological confidence is strong, you can actively participate in physical activities because you have the courage to face difficult tasks. This can promote success in other aspects of life, positive personality development, and smooth interpersonal relationships [13]. Ultimately, this state of mind enables a positive and confident life in general situations and school life.

Based on this context, the positive manifestation of basic psychological needs and high physical self-efficacy in adolescents can influence students’ feelings and attitudes toward PE [14] and facilitate a positive or change in mindset toward perceiving PE [15]. Additionally, basic psychological needs, physical self-efficacy, and attitudes in class are valuable and sufficient relative to stimulating an individual’s potential inner world [16,17]. As this can directly and/or indirectly affect students’ attitudes toward participating in class, investigating the relationship between basic psychological needs, physical self-efficacy, and attitudes during classroom experiences is a highly compelling task. This study contributes to the literature by showing how attitudes toward school life can be improved via physical activity in secondary PE classes. Therefore, this study’s purpose is to investigate the effects of basic psychological needs for South Korean middle-school PE on students’ physical self-efficacy and attitudes toward PE.

### 1.1. Relationships between Basic Psychological Needs, Physical Self-Efficacy, and Attitudes toward PE and Research Hypotheses

Basic psychological needs significantly affect physical self-efficacy, as they can maintain physical, psychological, and social health and comfort when the essential needs pursued by humans are fulfilled [4,18]. In studies on athletes; children; and middle- and high-school and college students, affirmative responses were found in terms of confidence in and expectations for being able to solve any task and affection with respect to one’s body if basic psychological needs were positive [15,16,17]. Therefore, our first research hypothesis (H1) is as follows:

**H1.** *Basic psychological needs for middle-school PE positively affect physical self-efficacy*.

Next, various studies on learning have identified the relationship between basic psychological needs and attitudes toward PE [19,20,21,22]. Basic psychological needs comprise three basic factors: autonomy, competence, and relatedness. Prior studies revealed that individual behavior and performance could vary depending on the degree to which these sub-factors are fulfilled or satisfied [4], and the degree of satisfaction is a major variable that can have a positive effect on proactivity, passion, and confidence [23]. In particular, it serves an important role in the relationship between students’ motivation to participate in class, emotions and behaviors toward class, and proactive attitudes toward class in the learning environment [14,24]. Consequently, the relationship between basic psychological needs and attitudes toward PE appears to be profound; if basic psychological needs are positive, such needs will also have positive effects on attitudes toward PE. Therefore, the second research hypothesis (H2) is as follows:

**H2.** *Basic psychological needs for middle-school PE positively affect attitudes toward PE*.

Finally, the relationship between physical self-efficacy and attitudes toward PE has been studied in terms of the behavioral intention, effort, and continuity of students’ PE participation in PE situations. These studies demonstrated a significant effect between the two variables [25,26,27,28]. Students’ beliefs of being able to perform a given task in PE can determine their lifelong attitudes toward PE [29,30]. This is because a positive experience is an important factor that affects attitudes [31]. Furthermore, various studies reveal that variables such as self-efficacy are important in that positive results drive students to be more interested in PE [32,33,34,35]. These results were more prominent in adolescents than in adults [36]. Ultimately, there is a profound relationship between the three variables: basic psychological needs, physical self-efficacy, and attitudes toward PE. Therefore, the third research hypothesis (H3) is as follows:

**H3.** *Physical self-efficacy in middle-school PE positively affects attitudes toward PE*.

### 1.2. Research Model

The hypotheses were based on the effects of the basic psychological needs of middle-school students during PE on physical self-efficacy and attitudes toward PE in the Republic of Korea. The hypothesis model is presented in Figure 1.

## 2. Materials and Methods

### 2.1. Participants

Middle-school students in Seoul and Gyeonggi Province—major population centers in South Korea—were recruited. A sample of 350 students was randomly sampled. Specifically, considering the characteristics of each region, Seoul was divided into four regions—east, west, south, and north—based on the Han River, and one school was selected for each of the 28 cities in Gyeonggi-do. Data from 32 schools were collected. After excluding 54 questionnaires with incomplete or missing responses, the data from 296 participants were analyzed. Using G*Power, a sample size of 118 was needed (actual power = 0.80); therefore, our current sample size is appropriate. Participants’ general characteristics are presented in Table 1.

### 2.2. Instruments

A 33-item structured questionnaire was administered to participants: 13 items for basic psychological needs, 10 items for physical self-efficacy, 8 items for attitude toward PE, and 2 demographic variables (sex and grade). For each question, the response was indicated as “strongly disagree” (1 point), “disagree” (2 points), “neutral” (3 points), “agree” (4 points), or “strongly agree” (5 points).

Basic psychological needs were assessed using items taken from Deci and Ryan [37] that were modified for the current study. The scale comprised three factors: four items for competence, four items for autonomy, and five items for relatedness. All were rated on a five-point Likert scale.

Physical self-efficacy was assessed using the questionnaire developed by Ryckman [38], which was modified for use in this study. The scale comprised two factors: four items for perceived physical competence and six items for physical self-presentation confidence. Again, all were scored on a five-point Likert scale.

Attitude toward PE was assessed using a questionnaire developed by Kenyon [39], which was modified for use in the current study as well. The scale comprised two factors: four items each for basic attitudes and social attitudes (scored on a five-point Likert scale).

### 2.3. Validity and Reliability of Instruments

The content validity of the questionnaire was evaluated by two Sociology of Sports professors and one researcher with a Ph.D. A confirmatory factor analysis (CFA) was performed using the maximum likelihood method. Cronbach’s α was calculated to measure internal consistency.

CFAs revealed the following fit indices: basic psychological needs (χ^2^ = 154, df = 57, TLI = 0.913, CFI = 0.937, SRMR = 0.057, and RMSEA = 0.087; Table 2); physical self-efficacy (χ^2^ = 49.1, df = 34, TLI = 0.937, CFI = 0.952, SRMR = 0.062, and RMSEA = 0.054; Table 3); and attitude toward PE (χ^2^ = 73.1, df = 19, TLI = 0.911, CFI = 0.940, SRMR = 0.039, and RMSEA = 0.079; Table 4). Cronbach’s αs ranged from 0.736 to 0.923, indicating good to excellent internal consistency. A good fit was determined at a value of 0.90 or more for TLI and CFI and a value of 0.08 or less for RMSEA [40].

### 2.4. Investigative Procedure

In this study, the researcher conducted visits to middle schools located in Seoul and Gyeonggi-do, central cities in Korea, and sought cooperation from the designated teachers in charge. Following the explanation of the survey’s purpose and the procedure for completing the questionnaire, participants used a self-assessment method to fill out the questionnaire. In cases where respondents found it difficult to complete the questionnaire independently, the researcher filled it out directly through a face-to-face survey. Data were collected from May to June 2022 (n = 350). The survey took 25 min to complete.

### 2.5. Data Processing

The data in this study underwent the following processing steps. First, a frequency analysis was performed using the SPSS 24.0 program. Second, confirmatory factor analy-sis and reliability verification (Cronbach’s α) were performed using the Jamovi 1.6.23 pro-gram to assess the validity and reliability, respectively [41,42,43]. Third, correlation analysis and standard multiple regression analysis were performed using the SPSS 24.0 program. The statistical significance level for this study was set at 0.05.

## 3. Results

### 3.1. Effects of Basic Psychological Needs on Attitudes toward PE and Academic Grit

#### 3.1.1. Correlations among Study Variables

The results of the correlation analysis between the factors, confirming the degree of satisfaction of discriminant validity for factors with established unidimensionality, are presented in Table 5. The correlations (r) among the variables varied between 0.253 and 0.798, indicating partially significant correlations. Since all correlation coefficients remained below 0.80, based on Kline’s criteria [44], discriminant validity was confirmed. In addition, all variables demonstrated values below the reference threshold of 0.80, indicating the absence of multicollinearity issues among independent variables [45].

#### 3.1.2. Effects of Basic Psychological Needs in Middle-School PE on Physical Self-Efficacy

Table 6 shows the results of the standard multiple regression analysis, which was performed to examine the effects of the basic psychological needs of middle-school PE on the components of physical self-efficacy: competence, autonomy, and relatedness.

The basic psychological needs of middle-school PE explained 47.5% (R^2^ adj. = 0.475) of the perceived physical competence component of physical self-efficacy. Moreover, the competence (β = 0.535) and autonomy (β = 0.320) components of basic psychological needs had significant effects on perceived physical competence (*p* = 0.000 and 0.004, respectively).

The basic psychological needs of middle-school PE explained 15.0% (R^2^ adj. = 0.150) of the physical self-presentation confidence component of physical self-efficacy. Moreover, the autonomy (β = 0.458) of basic psychological need components significantly affected physical self-presentation confidence (*p* = 0.001).

### 3.2. Effects of Basic Psychological Needs in Middle-School PE on Attitude toward PE

Table 7 shows the results of the standard multiple regression analysis, which was performed to examine the effects of the basic psychological needs of middle-school PE on the components of attitude toward PE: basic and social attitudes.

The basic psychological needs of middle-school PE explained 47.0% (R^2^ adj. = 0.470) of the basic attitude component for students’ attitudes toward PE. Moreover, the relatedness (β = 0.388) and autonomy (β = 0.317) components of basic psychological needs had significant effects on basic attitude (*p* = 0.000 and 0.004, respectively).

The basic psychological needs of middle-school PE explained 48.4% (R^2^ adj. = 0.484) of the social attitude component of students’ attitudes toward PE. Moreover, the competence (β = 0.214), autonomy (β = 0.349), and relatedness (β = 0.213) components of basic psychological needs had significant effects on social attitude (*p* = 0.019, 0.002, and 0.002, respectively).

### 3.3. Effects of Physical Self-Efficacy in Middle-School PE on Attitudes toward PE

Table 8 shows the results of the standard multiple regression analysis, which was performed to examine the effects of the physical self-efficacy of middle-school PE on the components of attitudes toward PE: basic and social attitudes.

The physical self-efficacy of middle-school PE explained 12.1% (R^2^ adj. = 0.121) of the basic attitude component of students’ attitudes toward PE. Moreover, the perceived physical competence (β = 0.258) and physical self-presentation confidence (β = 0.166) components of physical self-efficacy had significant effects on basic attitude (*p* = 0.004 and 0.059, respectively).

The physical self-efficacy of middle-school PE explained 18.8% (R^2^ adj. = 0.188) of the social attitude component of students’ attitude toward PE. Additionally, the perceived physical competence (β = 0.372) of physical self-efficacy significantly affected social attitude (*p* = 0.000).

## 4. Discussion

The objective of this study was to investigate the effect of basic psychological needs relative to middle-school PE on physical self-efficacy and attitudes toward PE. Participants were middle-school students located in Seoul and Gyeonggi-do in the Republic of Korea. Data from 296 students were analyzed.

First, basic psychological needs had a positive effect on perceived physical competence and physical self-presentation confidence, which were sub-variables of physical self-efficacy. Studies have demonstrated that well-being, effort, enjoyment, proactivity, and efficacy can be predicted if students can satisfy their basic psychological needs in PE, which can lead to positive results. Since the reason for the development of basic psychological needs theory is to verify individual stability, it is a theory that is deeply related to human behavior, such as school life satisfaction, class attitude, and physical efficacy [4,5,6,18]. This is partly consistent with past studies [14,15,16,17].

Notably, the findings demonstrated the importance of students’ inner psychological state in exhibiting physical self-efficacy in PE, and basic psychological needs affected students’ beliefs in their physical performance competence, confidence in performing tasks, and expectations that results will be favorable. Therefore, the psychological anxiety or discomfort of students in PE must be resolved. Additionally, anyone should be able to participate easily in PE to fulfill basic psychological needs. Students’ decision making in PE must be fostered [46], and we must create environments that enable unfettered networking for the desire to form peer groups with close friends [47]. Furthermore, help must be provided to students to ensure that they find PE enjoyable and exciting; regain their efficacy; and fully demonstrate their potential abilities [48].

Second, basic psychological needs had a positive effect on basic and social attitudes, which were sub-variables of attitudes toward PE. Various studies have noted that if students’ basic psychological needs are positive, they display positive attitudes in other classes at school, including PE, and it affects their concentration, confidence, achievement, satisfaction, and happiness [5,7,21].

Moreover, the satisfaction of the three factors comprising basic psychological needs enabled more positive attitudes, effort, persistence, and active participation in physical activity [49]. Basic psychological needs help prompt immersive behavior in students [50,51]. They can provide important clues for deciding how to perform actions that are required according to internal and/or external circumstances during the learning process in PE and prompt voluntary or positive behavior [48]. Basic psychological needs and PE attitudes are variables that produce positive results. Nevertheless, if the instructor controls students’ autonomy in PE and focuses on physical abilities and relative evaluations among students, negative results may ensue [51].

Students gain confidence through their actions. In turn, confidence inspires behaviors of achievement and consequently increases satisfaction. However, students’ psychological satisfaction was low because the grounds for the results of physical action were based on the outcome of the relative evaluation, with the objective of learning focused solely on the performance itself and designed to be result-oriented [52]. Ultimately, based on the understanding of various studies, to enhance students’ attitudes toward PE proactively and positively, it is important to let students work out tasks that are difficult to solve using physical functions and allow various experiences that engender positive results. When such processes are accumulated, they eventually affect psychological satisfaction.

Third, physical self-efficacy had a positive effect on students’ attitudes toward PE. The theory of physical self-efficacy is related to physical activity, PE, and exercise performance, as well as academic attributes such as school life, academic ability, and class attitude [25,26,27,28,29]. The findings that students with positive physical self-efficacy in PE put in considerable effort to achieve given tasks, endure difficulties for extended periods, and advance their attitudes toward learning are consistent with the study’s outcomes [53]. In particular, more positive results concerning the levels of confidence, proactivity, effort, and enjoyment in PE were found in co-ed classes comprising male and female students than in single-sex classes [54]. Ultimately, proactivity is demonstrated under circumstances in which students display physical abilities in front of others [55,56].

To increase physical self-efficacy, students’ interests and passions must be considered primarily. The experience of overcoming failures and persisting to succeed, even if one fails, must be encouraged. There is a profound relationship between basic psychological needs, physical self-efficacy, and attitudes toward PE. Teachers’ pedagogical strategies and instructional behaviors should be focused on students’ basic psychological needs and the promotion of physical efficacy and positive attitudes in PE.

In particular, the culture of middle and high schools in South Korea continues to have problems with deviant behavior such as school violence and extortion among students. This can foster fear and social resentment. To correct this social phenomenon, the Ministry of Education, Science and Technology expanded the number of hours of physical education classes to four hours per week to cultivate the “right personality” with respect to South Korean students [57]. Since this behavior is a problem that may appear in South Korean society in which competitiveness has been emphasized due to rapid development, it is necessary to further increase physical education hours. Similar cases may occur in countries with a similar culture, and these problems must be corrected via physical activity.

Therefore, teachers’ sense of the mission of physical education should be increased, and educational programs should be developed in which students can actively participate in physical education classes. ICT programs should be utilized, girls’ participation should be promoted, and simple team-specific leagues—rather than classes that end in general ball games—should be implemented.

## 5. Conclusions

In summary, this study investigated the effects of basic psychological needs in middle-school PE on students’ physical self-efficacy and PE attitudes. The basic psychological needs students significantly affected their physical self-efficacy and PE attitudes. It was found that students’ attitudes toward school life could be enhanced through their participation in physical activities during secondary PE classes.

This study had some limitations. First, we focused on the relationships between the study variables, which were analyzed through regression analyses; thus, differences between sexes and grades were not elucidated. Second, we only examined individual variables. Future studies should examine the influence of school-based PE on society while concurrently exploring the interplay of macro-level variables. Third, in regression analyses, sex, race, family characteristics, teacher characteristics, and school characteristics were not controlled for; however, these variables could affect attitudes toward PE and physical self-efficacy in students.

In addition, this study had the following limitations. First, the regional characteristics of Seoul and Gyeonggi-do were not reflected in the data collection. Thus, the results do not represent urban or rural areas, and they focus on simple random samples. Second, to cul-tivate the “right personality,” Korea increased the number of hours for middle-school PE classes to four hours per week. Accordingly, the Korean cultural context is reflected in the values of and attitude toward PE classes accordingly.

## 6. Suggestions for Future Studies

In future studies, to analyze the in-depth effects of sports activities, it is necessary to focus on the unique culture of sports in PE classes, set variables accordingly, and conduct in-depth research. In addition, a survey should be conducted to increase the satisfaction of sports activities via linkage to other academic fields. Therefore, we will need to find ways to increase our values, positive attitudes, and satisfaction with respect to the concepts of social and natural sciences.

## Figures and Tables

**Figure 1 healthcare-12-00091-f001:**
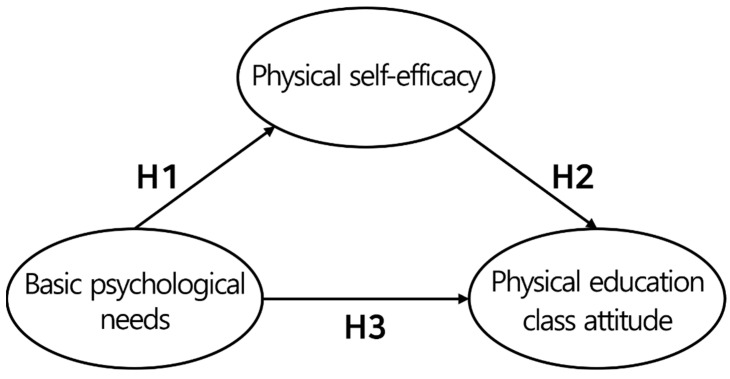
Study’s model.

**Table 1 healthcare-12-00091-t001:** Participants’ general characteristics.

Variable		n	%
Sex	Male	170	57.4
	Female	126	42.6
Grade level	6th	72	24.3
	7th	192	64.9
	8th	32	10.8

**Table 2 healthcare-12-00091-t002:** CFA and reliability of basic psychological needs.

Variable	Latent Variable	Measurement Variable	B	β	S.E.	t	AVE	CR	α
Basic psychological needs	Competence	a01	1	0.720			0.706	0.905	0.910
		a02	1.120	0.868	0.109	10.22 ***			
		a03	1.262	0.910	0.117	10.71 ***			
		a04	1.171	0.853	0.102	11.38 ***			
	Autonomy	b01	1	0.683			0.592	0.855	0.848
		b02	0.988	0.736	0.106	9.30 ***			
		b03	1.373	0.871	0.155	8.81 ***			
		b04	1.091	0.791	0.127	8.55 ***			
	Relatedness	c01	1	0.819			0.661	0.906	0.909
		c02	0.091	0.790	0.083	10.79 ***			
		c03	1.124	0.903	0.086	13.06 ***			
		c04	0.787	0.767	0.075	10.49 ***			
		c05	0.929	0.779	0.086	10.76 ***			

χ^2^: 154; df: 57; TLI: 0.913; CFI: 0.937; SRMR: 0.057; RMSEA: 0.087; *** *p* < 0.001.

**Table 3 healthcare-12-00091-t003:** CFA and reliability of physical self-efficacy.

Variable	Latent Variable	Measurement Variable	B	β	S.E.	t	AVE	CR	α
Physical self-efficacy	Perceived physical competence	a01	1	0.248			0.461	0.660	0.736
		a02	−2.811	−0.629	1.054	2.73 **			
		a03	−4.243	−0.819	1.520	2.79 **			
		a04	−4.237	−0.847	1.499	2.83 **			
	Physical self-presentation confidence	c01	1	0.584			0.273	0.668	0.749
		c02	1.089	0.654	0.211	5.16 ***			
		c03	0.369	0.209	0.174	2.12 *			
		c04	0.592	0.344	0.174	3.41 ***			
		c05	0.930	0.495	0.208	4.47 ***			
		c06	1.178	0.680	0.209	5.64 ***			

χ^2^: 49.1; df: 34; TLI: 0.937; CFI: 0.952; SRMR: 0.062; RMSEA: 0.054. * *p* < 0.05; ** *p* < 0.01; *** *p* < 0.001.

**Table 4 healthcare-12-00091-t004:** CFA and reliability of attitude toward PE.

Variable	Latent Variable	Measurement Variable	B	β	S.E.	t	AVE	CR	α
Attitude toward PE	Basic Attitude	a01	1	0.848			0.748	0.922	0.923
		a02	0.985	0.860	0.073	13.40 ***			
		a03	1.077	0.894	0.075	14.22 ***			
		a04	1.002	0.859	0.074	13.42 ***			
	Social Attitude	b01	1	0.751			0.578	0.844	0.840
		b02	0.981	0.769	0.102	9.62 ***			
		b03	1.159	0.872	0.109	10.60 ***			
		b04	0.958	0.632	0.126	7.60 ***			

χ^2^: 73.1; df: 19; TLI: 0.911; CFI: 0.940; SRMR: 0.039; RMSEA: 0.079; *** *p* < 0.001.

**Table 5 healthcare-12-00091-t005:** Correlations among basic psychological needs, physical self-efficacy, and attitudes toward PE.

	1	2	3	4	5	6	7
1. Competence	1						
2. Autonomy	0.749 ***	1					
3. Relatedness	0.626 ***	0.761 ***	1				
4. Perceived physical competence	0.675 ***	0.599 ***	0.419 ***	1			
5. Physical self-presentation confidence	0.315 ***	0.399 ***	0.253 **	0.460 ***	1		
6. Basic attitude	0.523 ***	0.644 ***	0.656 ***	0.334 ***	0.284 ***	1	
7. Social attitude	0.608 ***	0.671 ***	0.612 ***	0.431 ***	0.299 ***	0.798 ***	1

** *p* < 0.01; *** *p* < 0.001.

**Table 6 healthcare-12-00091-t006:** Standard multiple regression analysis for the effects of basic psychological needs on physical self-efficacy.

Variable	Perceived Physical Competence		Physical Self-Presentation Confidence		VIF
	β	t	β	t	
Competence	0.535	5.877 ***	0.052	0.451	2.317
Autonomy	0.320	2.928 **	0.458	3.293 ***	3.346
Relatedness	−0.159	1.717	−0.128	1.086	2.416
Adjusted R^2^ F	Adjusted R^2^ = 0.475 F = 45.333 ***		Adjusted R^2^ = 0.150 F = 9.621 ***		

** *p* < 0.01; *** *p* < 0.001.

**Table 7 healthcare-12-00091-t007:** Standard multiple regression analysis for the effects of basic psychological needs on attitudes toward PE.

Variable	Basic Attitude		Social Attitude		VIF
	β	t	β	t	
Competence	0.042	0.459	0.214	2.371 **	2.317
Autonomy	0.317	2.890 **	0.349	3.219 **	3.346
Relatedness	0.388	4.161 ***	0.213	2.310 *	2.416
Adjusted R^2^ F	Adjusted R^2^ = 0.470 F = 44.496 ***		Adjusted R^2^ = 0.484 F = 46.935 ***		

* *p* < 0.05; ** *p* < 0.01; *** *p* < 0.001.

**Table 8 healthcare-12-00091-t008:** Standard multiple regression analysis for the effects of academic grit on attitudes toward PE.

Variable	Basic Attitude		Social Attitude		VIF
	β	t	β	t	
Perceived physical competence	0.258	2.963 **	0.372	4.447 ***	1.269
Physical self-presentation confidence	0.166	1.902 *	0.128	1.530	1.269
Adjusted R^2^ F	Adjusted R^2^ = 0.121 F = 11.155 ***		Adjusted R^2^ = 0.188 F = 17.997 ***		

* *p* < 0.05; ** *p* < 0.01; *** *p* < 0.001.

## Data Availability

The data presented in this study are available upon request from the corresponding author.

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
