# Peer review of "Effects of Basic Psychological Needs on Physical Self-Efficacy and Attitudes toward PE in Korean Middle-School Physical Education"

_healthcare, 2023, doi:10.3390/healthcare12010091_

Round 1

Reviewer 1 Report

Comments and Suggestions for Authors

Author Response

As requested, the contents have been modified and supplemented.
Thank you for your comments and suggestions.

Reviewer 2 Report

Comments and Suggestions for Authors

Thank you for the opportunity to review your manuscript titled "Effects of Basic Psychological Needs on Physical Self-efficacy and Attitudes toward PE in Korean Middle School Physical Education." The study addresses a significant area in educational psychology and physical education, particularly in the context of adolescent development in South Korea. Your work provides valuable insights into how psychological needs influence students' attitudes and self-efficacy in physical education, a topic of great relevance in today's educational landscape.

Review for the Authors

Introduction: Thank you for submitting your manuscript titled "Effects of Basic Psychological Needs on Physical Self-efficacy and Attitudes toward PE in Korean Middle School Physical Education." Your research offers critical insights into the intersection of psychology and physical education in the context of adolescent development. It is a valuable addition to the field, particularly given its focus on the South Korean educational setting.

Suggestions for Improvement

Methodology:

  1. Sampling Method and School Selection:

    • Greater Detail on School Selection: The manuscript would benefit from a more comprehensive description of the school selection process. Specifically, clarify whether the schools were chosen based on their size, demographic characteristics, or academic performance. Were these schools representative of the typical educational environment in Seoul and Gyeonggi-do? This information is vital for assessing the study's ecological validity and the potential biases in your sample.
  2. Instrumentation and Adaptation:

    • In-depth Description of Modifications: Your study modifies existing questionnaires for your specific demographic. Please provide a detailed account of these changes. What were the criteria for adding or omitting certain items? How were the questions adapted to suit the cultural and educational context of South Korean middle school students? Were any terms or concepts altered for better comprehension by the younger audience? This level of detail will help in assessing the content validity of your instruments.
  3. Confirmatory Factor Analysis (CFA) Results:

    • Detailed Reporting of CFA Metrics: Expand on the confirmatory factor analysis by including specific metrics such as the Root Mean Square Error of Approximation (RMSEA), Comparative Fit Index (CFI), and Tucker-Lewis Index (TLI). What were the values obtained for these indices, and how do they compare to the commonly accepted thresholds for good fit? This detailed reporting will enhance the reader's confidence in the construct validity of your instruments and the robustness of your findings.
  4. Discussion:

    1. Linking Findings to Broader Educational Implications:

      • Contextualizing Results: While your discussion effectively ties the findings to the importance of psychological needs in PE, a deeper connection to broader educational theories and practices would be beneficial. How do these results align with, or challenge, existing theories on adolescent learning and development in physical education?
      • Integration with International Research: Considering the cultural specificity of your study, it would be enriching to discuss how your findings compare to similar studies in different cultural or educational settings. This comparative analysis can provide valuable insights into the universality or specificity of your findings.
    2. Practical Applications:

      • Concrete Recommendations for PE Curriculum: Expand your discussion to offer more concrete, actionable recommendations for PE teachers and curriculum developers. For instance, specific strategies for integrating autonomy, competence, and relatedness into PE activities could be provided, based on your findings.

    Limitations:

    1. Sample Representativeness:

      • Addressing Potential Bias: More attention should be paid to discussing how the selection of schools and students may have introduced bias. For example, if schools in urban areas were overrepresented, how might this affect the applicability of your findings to rural schools?
    2. Instrumentation Limitations:

      • Cultural Appropriateness of Instruments: While the modifications made to the questionnaires are noted, the limitations these changes may introduce should be discussed. For example, were the scales originally developed in a different cultural context, and if so, how might this affect their validity in a South Korean setting?

Author Response

(The authors gave the same response as above.)

Reviewer 3 Report

Comments and Suggestions for Authors

This is a topic that has a long history, it is not a current topic. The work as a whole is adequate and as the results show, everything is correct, where there is a relationship between the study variables.

Some aspects of improvement would be to introduce a paragraph after hypothesis three as in the case of hypotheses one and two (p.3).

It would be advisable to specify how the five-point Likert scale is scored, if 1 scores less than 5 or how it is considered within the analyses.

I consider tables 2, 3 and 4 that if they come with a brief prior explanation it would be more organized, because all the information is indicated in an initial paragraph and then all the information in the tables is presented continuously without any explanation, it would be It is appropriate to break down all that information for better reading comprehension.

Expand the conclusions because very synthetic information is presented and put the limitations in a new section, not within the conclusions section.

As it is a topic in which a long research journey, the references are mostly longer than 5-10 years, which is recommended, for this reason an update of more recent studies in both the citations and the references would be appropriate.

Author Response

(The authors gave the same response as above.)

Round 2

Reviewer 2 Report

Comments and Suggestions for Authors

To the Authors of Manuscript,

Upon review of your revised manuscript, I am pleased to note that the concerns previously raised have been thoroughly addressed. The introduction section now effectively underscores the significance of physical education (PE) and its comprehensive benefits for students, especially during the pivotal developmental stage of adolescence. Additionally, the connection you have drawn between physical self-efficacy and students' attitudes towards PE is both clear and compelling.

To further enhance the manuscript and emphasize its relevance to ongoing scholarly discussions, I recommend incorporating additional contemporary studies within the introduction. Recent research can provide updated insights and emphasize the present-day relevance of your study. I suggest citing works such as those by Longobardi et al. (2022), which delve into the perceptions of teachers regarding the physical appearance of primary school students and its correlation with the quality of student-teacher relationships and peer popularity. Moreover, the study by Ouyang et al. (2020) examines the impact of sports participation on body image, self-efficacy, and self-esteem in college students, and Longobardi et al. (2023) explore the implications of teacher sentiments on physical appearance and bullying victimization. These references can significantly contribute to the narrative of your study, offering a modern context that aligns with your research objectives.

Integrating these recent references will not only reinforce the importance of your study but also align it with the current academic discourse on the psychological effects of PE on students. I recommend that these studies be interwoven with the existing content in your introduction to provide a strong foundation for your work.

In summary, the revisions made to the manuscript are satisfactory and, with the addition of the suggested recent references, it will be well-positioned for publication.

Here are some suggestions:

Longobardi C., Mastrokoukou, S., & Fabris, M. A. (2023). Teacher sentiments about physical appearance and risk of bullying victimization: the mediating role of quality of student–teacher relationships and psychological adjustment. Front. Educ. 8:1211403. https://doi.org/10.3389/feduc.2023.1211403

Longobardi, C., Settanni, M., Berchiatti, M., Mastrokoukou, S., & Marengo, D. (2022). Teachers’ sentiment about physical appearance of primary school students: Associations with student–teacher relationship quality and student popularity among classroom peers. Social Psychology of Education, 1-21. https://doi.org/10.1007/s11218-022-09749-9.

Ouyang, Y., Wang, K., Zhang, T., Peng, L., Song, G., & Luo, J. (2020). The influence of sports participation on body image, self-efficacy, and self-esteem in college students. Frontiers in psychology10, 3039.

Author Response

(The authors gave the same response as above.)
